# Overcoming chemotherapy resistance in low-grade gliomas: A computational approach

Thibault Delobel[1,2�], Luis E. Ayala-Hernández[1,3�], Jesús J. Bosque[1]*, Julián Pérez-Beteta[1], Salvador Chulián[1,4], Manuel García-Ferrer[5], Pilar Piñero[5], Philippe Schucht[6], Michael Murek[6], Víctor M. Pérez-García[1]

**1** Department of Mathematics, Mathematical Oncology Laboratory (MOLAB), University of Castilla-La Mancha, Ciudad Real, Spain, **2** Sorbonne Université, Paris, France, **3** Departamento de Ciencias Exactas y Tecnología Centro Universitario de los Lagos, Universidad de Guadalajara, Lagos de Moreno, Mexico, **4** Department of Mathematics, Universidad de Cádiz, Biomedical Research and Innovation Institute of Cádiz (INiBICA), Hospital Universitario Puerta del Mar, Cádiz, Spain, **5** Department of Radiology, Virgen del Rocío University Hospital, Seville, Spain, **6** Department of Neurosurgery, Inselspital Bern and University Hospital, Bern, Switzerland

उ These authors contributed equally to this work.
\* jesus.bosque@uclm.es

**Data Availability Statement:** All relevant data are within the manuscript and its Supporting information files. All code used for running experiments, model fitting, and plotting is available

## Abstract

Low-grade gliomas are primary brain tumors that arise from glial cells and are usually treated with temozolomide (TMZ) as a chemotherapeutic option. They are often incurable, but patients have a prolonged survival. One of the shortcomings of the treatment is that patients eventually develop drug resistance. Recent findings show that persisters, cells that enter a dormancy state to resist treatment, play an important role in the development of resistance to TMZ. In this study we constructed a mathematical model of low-grade glioma response to TMZ incorporating a persister population. The model was able to describe the volumetric longitudinal dynamics, observed in routine FLAIR 3D sequences, of low-grade glioma patients acquiring TMZ resistance. We used the model to explore different TMZ administration protocols, first on virtual clones of real patients and afterwards on virtual patients preserving the relationships between parameters of real patients. In silico clinical trials showed that resistance development was deferred by protocols in which individual doses are administered after rest periods, rather than the 28-days cycle standard protocol. This led to median survival gains in virtual patients of more than 15 months when using resting periods between two and three weeks and agreed with recent experimental observations in animal models. Additionally, we tested adaptive variations of these new protocols, what showed a potential reduction in toxicity, but no survival gain. Our computational results highlight the need of further clinical trials that could obtain better results from treatment with TMZ in low grade gliomas.

## Author summary

Low-grade gliomas are incurable brain tumors that originate from glial cells, the cells that provide physical and chemical support to neurons. Patients typically receive

on a GitHub repository at https://github.com/ThibDelo/LGG_resistance. We have also used Zenodo to assign a DOI to the repository: https://doi.org/10.5281/zenodo.8387519.

**Funding:** This work has been partially supported by the Spanish Ministerio de Ciencia e Innovación, grant numbers PID2019-110895RB-I00 (MCIN/AEI/10.13039/501100011033) and TED2021-132318B-I00, and by Junta de Comunidades de Castilla-La Mancha (grant SBPLY/21/180501/000145), all to V.M.P-G. The funders had no role in study design, data collection and analysis, decision to publish, or preparation of the manuscript.

**Competing interests:** The authors have declared that no competing interests exist.

temozolomide (TMZ) chemotherapy as part of the standard treatment, but eventually develop resistance to the drug, what constitutes an important therapeutic challenge. We developed a mathematical model to explore novel TMZ delivery protocols that could improve survival and reduce toxicity. These are grounded in the reduction of the persister population, a recently discovered glioma cell type that reversibly changes to a quiescent state to resist insults. We measured tumor volume from longitudinal imaging studies performed in patients that showed resistance to treatment and used those measurements to validate our mathematical model. We proposed a general scheme of TMZ administration that consisted in delivering isolated doses with long resting periods, in contrast to classical cycle delivery with higher concentration of doses. Computational simulations modified schemes showed a benefit in survival with reduced toxicity. Our findings could guide biological experiments aimed at improving overall survival and quality of life for patients with low-grade gliomas.

## Introduction

Gliomas are a heterogeneous group of primary brain tumors that originate from glial cells, the supporting cells of neurons. They represent 24% of all central nervous system (CNS) tumors, and are therefore the most common primary brain tumors of the CNS [1]. Gliomas are classified into three categories depending on their severity: benign (WHO grade 1), low-grade (WHO grade 2) and high-grade (WHO grade 3–4). Low-grade gliomas (LGG) usually harbor mutations in the gene encoding isocitrate dehydrogenase 1 (IDH1) and show a distinct behavior from their high-grade counterparts [2]. They are slow growing and most often occur in young adults. Due to their infiltrative nature, the tumor cells invade the surrounding brain, making LGGs usually incurable with near-systematic recurrence, even after total resection [3]. With time, LGGs acquire new mutations, and eventually evolve into higher grade, malignant, and much more aggressive tumors [4]. Nevertheless, survival can exceed 10 years with a 5-year survival rate estimated at 50% [1].

The optimal management of LGGs remains unclear [5]. After first-line surgery, the patient usually receives chemotherapy, radiotherapy, or radiochemotherapy [6]. Temozolomide (TMZ) is one of the most widely used antitumor agents against LGGs, mainly due to its effectiveness and relatively low toxicity. However, TMZ is not exempt from one of the major problems facing anti-cancer therapies: drug resistance. Many mechanisms can explain the development of such resistance, but it is now becoming increasingly clear that a recently discovered cell type, persistent cells, or persisters, are key to explaining the development of such acquired resistance [7–11]. These cells have a slow or dormant metabolism, what makes them tolerant to drugs. Under treatment-induced selection pressure, these cells are able to evolve towards a resistant phenotype, but can also revert to a sensitive phenotype in the absence of drug exposure. Several recent studies have shown that persistent cells may explain the development of TMZ resistance in glioblastoma [12–14]. Experiments conducted by Segura et al. [13] in slow-growing glioblastoma cell lines and orthotopic glioblastoma mouse models have shown that spacing out individual TMZ doses increased cell viability/mouse overall survival and decreased the amount of resistance-associated factors. This suggests that spacing individual doses may provide more time for persisters to revert to a sensitive state, and thus delay the emergence of resistance. Interestingly this type of study has never been conducted in LGGs, mainly due to the lack of availability of good experimental models of the disease. Nevertheless, due to their similarity and common precursor cell, it is reasonable to expect that the results

obtained in glioblastomas regarding the existence of persisters can be translated to the case of LGGs [13]. Thus, it is crucial that the populations of persistent cells are taken into account in the modeling of the disease and therapies are adapted consequently.

In clinical trials, the identification of an optimal protocol is often difficult. Indeed, the choice between different therapies, drugs concentration and their administration schedule imposes a large number of potential combinations, often too many to be tested by classical empirical approaches. This is particularly true for LGG: here, the rarity of the disease makes recruitment of patients into trials difficult, and long natural overall survival lead to very long trial duration times. In this context, mathematical modeling and computational simulations are proving to be powerful tools to explore new clinical protocols, via a virtual, low-cost, fast and patient-free approach [15, 16]. A mathematical model is based on strong assumptions, and is not intended to perfectly reproduce the real world. The purpose of a model is to explore certain hypotheses by placing them in an abstract and simplified context. This kind of approach can generate ideas or "proofs of concept" to guide clinical research.

This work has two goals. The first one is to build a mathematical model able to describe the macroscopic growth of LGGs under the influence of TMZ chemotherapy. The second one is to use this model to implement a virtual clinical trial that identifies an improved TMZ protocol postponing the emergence of acquired TMZ resistance, while controlling tumor growth and reducing treatment toxicity. Several LGGs models have been previously developed [17–22] but, to our knowledge, none of them have included the emergence of resistance considering the role of persister cells. Here we developed such a model and validated it by fitting longitudinal volumetric data from longitudinal MRI data of LGG patients showing evidence of acquired TMZ resistance. For each patient, we obtained a set of parameters describing the tumor behavior, what allowed us to create virtual copies of each real patient. Using those virtual twins, we conducted simulations to explore new TMZ administration schemes. We found that spacing out the same amount of TMZ in the form of individual doses lead to a delay in the emergence of resistance with respect to the 28-days cycle standard protocol. Finally, to extend our results we conducted a virtual clinical trial generating a cohort of in silico patients with variable values of the parameters. Our results suggested that dose protocols with 14 or 21 days intervals between individual doses significantly delay the emergence of resistance in comparison to the conventional administration protocol.

## Materials and methods

### Ethics statement

The study was approved by Kantonale Ethikkommission Bern (Bern, Switzerland), with approval number: 07.09.72, and by the Institutional Review Board of Hospital Virgen Macarena y Virgen del Rocío with approval number 2158-N-19. Written informed consent was obtained for all participants in the study.

### Formulation of the mathematical model

Rabé et al. [12] studied the acquisition of resistance of human glioma cells to TMZ performing in vitro and in vivo longitudinal experiments. Their results recognized three main cell types with disparate behavior: drug sensitive, drug-tolerant and fully resistant cells. *Drug-tolerant* state is caused by epigenetic alterations due to TMZ exposure, what produce a change in gene expression but do not initially affect the DNA sequence. Since there are no mutations, this cellular state can be reversed, i.e. if TMZ exposure is ceased, the drug-tolerant cells become sensitive again. These cells exhibit a slow proliferation rate allowing them to survive treatment and then give rise to permanently resistant cancer cells if exposure to the drug persists in time.

Drug-tolerant cells share common antibiotic tolerance properties observed in bacteria, thus they are also known as persistent cancer cells or persisters, replicating the term used for bacteria [9]. However, precise characterization of this particular population remains a major challenge in cancer biology (see e.g. [10, 14] for more details). Fig 1A shows schematically the most important cell types and a general framework of the resistance acquirement process, inferred from Rabé et al. work [12].

The treatment is also associated to an increase in the tumor aggressiveness. Together with resistance to treatment, TMZ exposure induces the proneural to mesenchymal phenotypic transition, in which cells change to a more aggressive state that is associated to a poor prognosis [13, 23]. Actually, this transition is a step towards resistance acquisition, and the two effects represent two sides of the same coin. The velocity of diametric expansion (VDE) of low-grade gliomas, estimated from the evolution of T2-weighted MRIs over time, was assessed by Pallud et al. [24]. They found that, in most patients, the VDE before treatment (mean 5.9 mm/year)

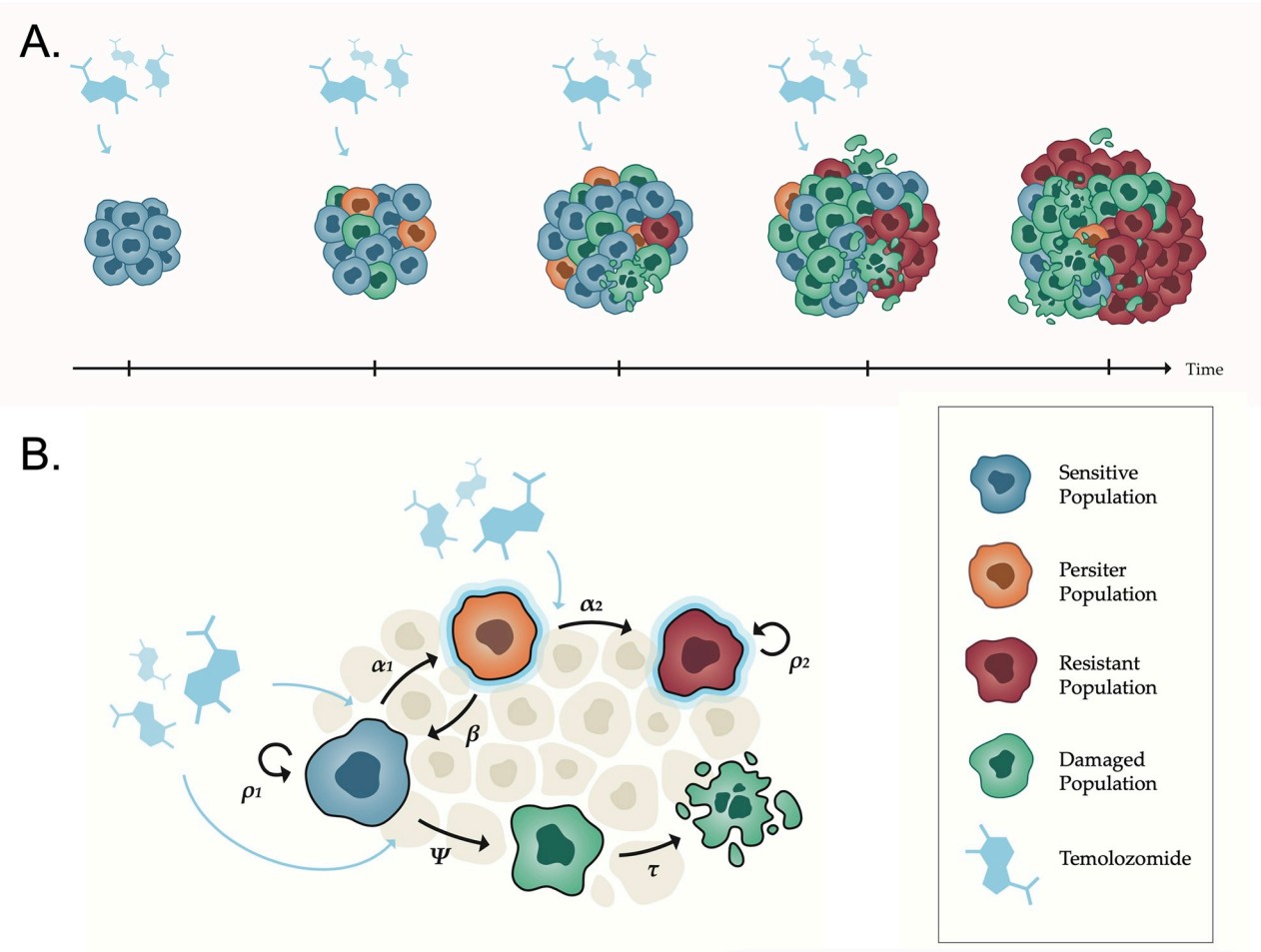

**Fig 1. Visualization of the resistance acquisition process and its translation to biological interactions between populations.** (A) At the beginning of TMZ treatment, there are only sensitive cells (blue). When TMZ is administered, some sensitive cells enter a persistent state (orange) and others are damaged (green). Under continuous administration of TMZ, cells acquire a fully resistant phenotype (red). (B) Diagram of interactions between the different populations as modeled by Eqs (1–6). Sensitive cells proliferate at a rate $\rho_1$. When TMZ is administered, some sensitive cells are damaged at a rate $\psi$ and enter the persistent state at a rate $\alpha_1$ under the exposure to TMZ. Persister cells can return to a sensitive state at a rate $\beta$ or give raise to a fully resistant phenotype at a rate $\alpha_2$ if TMZ exposure continues. Damaged cells die due to mitotic catastrophe at a rate $\tau$, while resistant cells grow at a rate $\rho_2$. Assumptions are based on biological experiments and clinical observations of patients diagnosed with LGGs.

was lower than the VDE after the treatment (mean 7.8 mm/year) when the tumor relapses, pointing to a higher proliferation rate of cells exposed to TMZ.

We developed a mathematical model of LGG growth and response to TMZ taking into consideration all the previous experimental results and clinical observations. The model was based on ordinary differential equations (ODE) and describes the evolution in time of the volumes occupied by the following four well-mixed tumor cell populations:

- **Sensitive cells** ($V_S$): Cells in which TMZ has an effect, which would be assumed to be proportional to the concentration of drug $E$ [25, 26]. We will assume that the generation of damage in the cells due to the alkylating effect of the drug has a rate $\psi$, and the promotion of the emergence of reversible persistent behavior has a rate $\alpha_1$. Additionally, cells in this compartment proliferate at a rate $\rho_1$ [27–29].

- **Damaged cells** ($V_D$): There are different modes of LGG cell death due to TMZ cytotoxic action [30]. We considered that the one that best explains the LGG clinical response to TMZ observed in patients by Ricard et al. [31] is the mitotic catastrophe. This implies a delayed effect that is noticeable months after the application of the treatment [32]. Thus we assume that cells exposed to TMZ first transit into this compartment and then decay to dead cells with characteristic time $\tau$.

- **Persister cells** ($V_P$ and $V_{PI}$): These are quiescent cells due to the exposure of sensitive cells to the TMZ. There are two stages of the process of persister stabilization before the sensitive cells can reach the resistant state: First, sensitive cells enter the intermediate and temporary persistent state $V_{PI}$ due to the exposure to high concentrations of drug $E$. Having been exposed to TMZ, these initial persisters can stabilize to a fully persistent phenotype $V_P$, which happens after the administration of the drug, when the concentration $E$ is low. This behavior is ruled by the transit function $f(E) > 0$, what has appreciable values only for very low values of $E$. If the drug concentration $E$ returns to high values due to a subsequent dose, the persister $V_P$ can become fully resistant $V_R$ with a rate $\alpha_2$. In the absence of drug exposure, these cells can also revert to a sensitive phenotype with a rate $\beta$.

- **Resistant cells** ($V_R$): Cells that stop being sensitive to TMZ due to a continuous exposure of persistent cells $V_P$ to the drug $E$. The transformation of sensitive cells to resistant cells is associated to the proneural to mesenchymal transition, in which cells acquired a more aggressive phenotype [13]. Consequently, they proliferate with a rate $\rho_2$ that should obey $\rho_2 > \rho_1$ according to Pallud's observations [24].

In addition to the those cellular populations, the normalized concentration of the drug $E$ is tracked in the model. TMZ is known to exhibit linear kinetics [33], i.e. the instantaneous rate of change in drug concentration depends only on the current concentration [34]. Thus, $E(t)$ can be interpreted as the effect of TMZ with a clearance rate $\lambda$. As a consequence, $0 \leq E(t) \leq 1$, where 1 is the largest possible effect and 0 corresponds to no effect.

The overall scheme of the previously described agents and their interactions is shown in Fig 1B. The time evolution of the described populations subject to the described interplay can be mathematically modeled by the following ODE system:

$$\frac{dV_S}{dt} = \overbrace{\rho_1 V_S}^{\text{proliferation}} \underbrace{-\psi V_S E}_{\substack{\text{to damaged} \\ \text{due to the drug}}} \underbrace{-\alpha_1 V_S E}_{\substack{\text{to persister} \\ \text{due to the drug}}} \overbrace{+\beta V_P}^{\text{from persister}}, \tag{1}$$

$$\frac{\mathrm{d}V_D}{\mathrm{d}t} \quad = \quad \overbrace{\psi V_S E}^{\text{from sensitive}} \underbrace{-\tau V_D}_{\text{death}},\tag{2}$$

$$\frac{\mathrm{d}V_{PI}}{\mathrm{d}t} \quad = \quad \overbrace{\alpha_1 V_S E}^{\text{from sensitive}} - V_{PI} f(E),\tag{3}$$

$$\frac{\mathrm{d}V_P}{\mathrm{d}t} \quad = \quad V_{PI} f(E) \underbrace{-\alpha_2 V_P E}_{\text{to resistant}} \quad \underbrace{-\beta V_P}_{\text{to sensitive}},\tag{4}$$

$$\frac{\mathrm{d}V_R}{\mathrm{d}t} \quad = \quad \overbrace{\rho_2 V_R}^{\text{growth}} \quad \overbrace{+\alpha_2 V_P E}^{\text{from persister}},\tag{5}$$

$$\frac{\mathrm{d}E}{\mathrm{d}t} \quad = \quad -\lambda E,\tag{6}$$

where all variables and parameters have been previously described. The initial conditions for Eqs (1–6) are $V_S(0) = V_{S0}, V_D(0) = V_{PI}(0) = V_P(0) = V_R(0) = E(0) = 0$, with $V_{S0}$ being the value of the first observation $V_O$ for each patient.

The activation function $f(E)$ was chosen to reproduce the following biological behavior in the simplest mathematical way: The exposure to TMZ promotes the transition of sensitive cells to persisters, but it also leads the transformation of persisters in resistant cells; however, the exposure to TMZ has to be reiterated for the last transition to happen, therefore, our model must prevent recently created persister cells from changing to resistant in the same dose application. In order to reproduce this, we consider an intermediate persister population and an activation function that should be close to zero for large values of the effect $E$ and positive for low values of $E$ to allow the transition from $V_{PI}$ to $V_P$. Therefore, we used a classical sigmoid pass function

$$f(E) = 7.5 \left(1 - \tanh\left(\frac{E - 0.01}{0.01}\right)\right).\tag{7}$$

Notice that this modeling approach is restricted to the case of TMZ, which is administered in pills, with no less than a day between subsequent administration, and has a half-life of approximately 2 hours. Therefore, the use of an activation function would fail if we were considering a drug subject to continuous uptake over a prolonged time period. Additionally, it would be possible to achieve this desired delay of transition times from $V_{PI}$ to $V_P$ by using a series of intermediate compartments, $V_{P0}$, $V_{P1}$, ..., $V_{Pn}$, $V_P$, what would provide more generality for the case of treatments that are administered continuously. See supporting information S2 File for more details.

## LGG patients imaging data

Longitudinal imaging data from histology confirmed LGG patients who had been treated with TMZ were used to assess the goodness of fit of the ODE model, infer the values of the parameters and analyze modifications in the treatment scheme.

Data were provided by three hospitals: Bern (Switzerland), Virgen Macarena (Seville, Spain) and Virgen del Rocío (Seville, Spain). All patients signed informed consent and the

study had been approved by the Institutional Review Boards of the hospitals. The authorization codes were: 07.09.72 (Bern University Hospital), 2158-N-19 (Virgen Macarena University Hospital) and 2158-N-19 (Virgen del Rocío University Hospital).

Data from a total of 91 patients were initially considered. Raw FLAIR 3D MRI studies of patients from Virgen Macarena University Hospital and Virgen del Rocío University Hospital was processed as explained below. Already processed volumetric longitudinal data from Bern University were used.

Patients that showed evidence of acquired resistance to the treatment with TMZ were included in the study. This was assessed as a volumetric regrowth under TMZ treatment after a previous initial response to the treatment. Patients not responding to TMZ were excluded from the study. Patients who underwent surgery or other interfering therapies in the period of study were excluded.

A total of 7 patients with grade-II gliomas (all male; median age 49 years with range 37–59 years) were selected. The final cohort consisted of four patients with oligodendroglioma and three patients with diffuse astrocytoma. Individual patient characteristics, longitudinal volumetric data and information related to TMZ treatment can be found in S1 File.

## Tumor volume measurement

Patients from Virgen Macarena and Virgen del Rocío University Hospitals had fluid attenuated inversion recovery (FLAIR) 3D MRI sequences available. Tumor volume calculation was performed following the procedure described in [35]. Digital Imaging and Communications in Medicine (DICOM) images were loaded into Matlab software (R2022b, The MathWorks, Inc., Natick, Massachusetts) and were semi-automatically delineated using a gray-level threshold to identify the tumor region. A slice by slice manual correction of each segmentation was then done by a research team member (T.D.) under the supervision of an image expert with six years of experience (J.P.B).

Once tumor regions were identified and delineated, the tumor volume was computed by counting the number of voxels in the segmented tumor and multiplying by the individual voxel volume. For comparison purposes, we also used the ellipsoid approximation method, which uses three orthogonal linear measures in the tumor [36, 37]: the largest tumor diameter $D_1$ in the axial plane was measured and the second axis was selected perpendicular to it ($D_2$); the third measure was obtained as the largest diameter $D_3$ in the sagittal, or equivalently, coronal planes (see S1 Fig). The total tumor volume was obtained by applying the formula $V = (D_1 \times D_2 \times D_3)/2$. Volumetric growth data from MRI scans of patients in Bern Hospital was obtained exclusively by the ellipsoid approximation by using diameters on successive T2/FLAIR sequences.

The first methodology yields more precise volume measurements at the cost of a greater time effort. For a fixed set of tumors, both methodologies were compared. An average difference of 18% was found, therefore, we used this value as the uncertainty in the segmentation and used an 18% error bar for the volume data.

## Modeling of the TMZ effect and pharmacokinetics

Peak TMZ concentration is reached about one hour after administration [33], what is very fast compared to the time required to observe considerable tumor growth. Therefore, tumor size can be considered to be constant between the time of drug administration and the time of peak concentration. Thus, the time to reach peak concentration was considered instantaneous. The evolution of volumes and drug concentration at the time $t_{T,i}$ of administration of a dose $i$,

with $i = 1, 2, \ldots, M$, being $M$ the total number of doses, was taken as

$$V_S(t_{T,i}) = V_S(t_{T,i}^-),$$ (8)

$$V_D(t_{T,i}) = V_D(t_{T,i}^-),$$ (9)

$$V_{PI}(t_{T,i}) = V_{PI}(t_{T,i}^-),$$ (10)

$$V_P(t_{T,i}) = V_P(t_{T,i}^-),$$ (11)

$$V_R(t_{T,i}) = V_R(t_{T,i}^-),$$ (12)

$$E(t_{T,i}) = E(t_{T,i}^-) + E_0,$$ (13)

with $t_{T,i}^-$ being the time just before the administration of the $i$ chemotherapeutic dose and $E_0$, the effect produced by the peak concentration of the given dose. TMZ is eliminated with a mean half-life of $t_{1/2} \approx 2$ h [33], therefore, the corresponding clearance rate is $\lambda = 24 \cdot \ln(2)/2 = 8.32$ day$^{-1}$.

## Estimation of patient-specific parameters

The specific behavior of each patient was captured in our model by the parameters $\rho_1, \psi, \alpha_1, \beta, \tau, \alpha_2$ and $\rho_2$. LGGs, as other types of brain tumors, are characterized by a range of inter-patient variability, therefore, different patients are expected to be represented by different individual values of these parameters.

The specific set of values providing the better fit of each patient evolution and response to TMZ was calculated by least mean squares. For each particular patient, we minimized the root mean squared error between total tumor volume in the model ($V(t) = V_S(t) + V_D(t) + V_{PI}(t) + V_P(t) + V_R(t)$) and in the longitudinal data ($V_{O,j}$ at times $t_{O,j}$, with $j = 1, \ldots, N$ being the consecutive clinical observations). During this process we performed the fit of all the parameters using the data of the overall volume evolution in time, that is, we did not distinguish different regimes in which independent parameters could be identified. The fits were implemented as a numerical optimization process. To ensure the robustness of the fit, several initial random seeds within the prescribed bounds were used. This raises the probabilities that the result of the optimization process is a global minimum (best fit), rather than a local minimum. From those different fits corresponding to different seeds we selected the fit showing a smaller mean squared error.

Further information from biological literature was used to constrain the values and relationships between parameters. It is known that one TMZ dose is enough to cause the emergence of persisters [12], however, more than one dose is needed to give rise to fully resistant cells. Under exposure to TMZ, the transition to the persister state from the sensitive cells is less demanding than the full transition from persister to resistant. Therefore, we forced the rates of generation of persistent and resistant cells to comply $\alpha_2 < \alpha_1$. As to the rate $\beta$ at which persisters return to sensitive, we set $\beta = 0.1$ d$^{-1}$ after the preliminary fits yielded a value $\beta \approx 0.1$ d$^{-1}$ for all the patients (see S3 File). This value is in full agreement with the persistent state duration of glioblastoma cells obtained from in vitro and in vivo experiments [12, 13].

All calculations and simulations were performed in Python version 3.9.7 (Python Software Fundation). The model was solved using the `odeint` function and fitted to the longitudinal

volumetric data using the `minimize` function with the Nelder-Mead method [38]. Both functions are included in version 1.7.1 of the SciPy package [39].

## Evaluation of TMZ administration protocols tested on virtual clones of patients

We intended to construct different TMZ application protocols that deferred the development of resistance and led to potentially better overall survival (OS). Our rationale was reducing the persister population and delaying the induction of resistance by spacing out the administration of TMZ doses.

We tested the gain of OS of the new protocols in comparison to the standard protocol (denoted as C28), which consists of cycles of 28 days with TMZ administration on days 1 to 5 and a break on days 6 to 28. The new protocols were organized in individual doses (ID) spaced out in weeks for convenient clinical implementation. They were distinguished by the number of days between each single doses. For instance, ID7 was a protocol in which the patient would receive doses with a resting time of 7 days (Fig 2).

The effectiveness of the different ID protocols was tested on virtual copies of the patient, characterized by their fitting parameters. Each ID protocol was simulated with the cumulative number of TMZ doses received during its real treatment. The OS of the virtual patients was defined as the time at which the simulated volume $V(t)$ reached a critical value, which was set randomly for each patient by sampling a Gaussian distribution with mean 280 $cm^3$ [18] and variance 20 $cm^3$.

## Generation of virtual patients

In our mathematical model, each patient is characterized by a set of parameter values that includes initial tumor volume, patient-specific parameters, number of TMZ cycles received and fatal volume. Therefore, virtual patients can be generated by randomly assigning these sets of values using different statistical methods.

First, the pre-chemotherapy tumor volume distribution was determined. We measured the volumetric data from MRI images of 32 patients at the time of diagnosis and determined the empirical probability distribution that they follow using `KernelDensity` function with a gaussian kernel from Scikit-Learn Python package, version 0.24.2 [40]. The obtained empirical

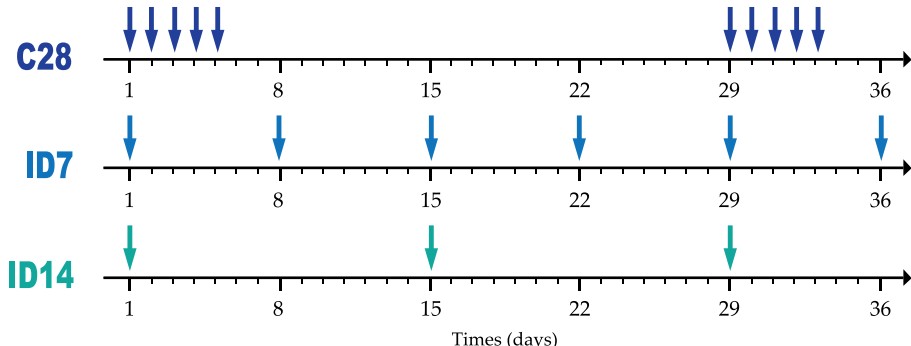

**Fig 2. TMZ administration protocols.** Each arrow represents an oral administration of TMZ. The C28 protocol (C for cyclic) consists of a given number of cycles of one dose per day for 5 consecutive days, followed by 23 days of rest. The ID protocol (ID for individual dose) consists of spacing the individual doses by a given number of weeks, in this example one (ID7) or two (ID14).

**Table 1. Patient-specific parameter values.**

| Patients | $V_S$ proliferation $\rho_1$ (×10$^{-4}$) | $V_R$ proliferation $\rho_2$ (×10$^{-3}$) | $V_D$ death $\tau$ (×10$^{-3}$) | $V_S$ damage $\psi$ (×10$^{-1}$) | $V_S \rightarrow V_{PI}$ $\alpha_1$ (×10$^{-1}$) | $V_P \rightarrow V_R$ $\alpha_2$ (×10$^{-2}$) |
|---|---|---|---|---|---|---|
| 1 | 12.2 | 5.54 | 5.82 | 9.79 | 5.13 | 8.9 |
| 2 | 6.65 | 1.98 | 2.09 | 2.9 | 2.2 | 4 |
| 3 | 12.6 | 4.35 | 2.89 | 5.33 | 3.5 | 6 |
| 4 | 8.26 | 10.2 | 8.9 | 3.96 | 4.98 | 9.9 |
| 5 | 6.24 | 1.796 | 1.69 | 1.9 | 2.55 | 5.3 |
| 6 | 9.4 | 7.25 | 9.64 | 3.9 | 2.68 | 7.5 |
| 7 | 5.7 | 1.73 | 2.57 | 3.5 | 3.88 | 7.2 |

Values for the different rates that parametrize the model for each fit presented on Fig 3. All units are in day$^{-1}$.

distribution fitted is depicted in S2A Fig. Afterwards, the initial tumor volumes for virtual patients were assigned by generating random values sampled from the empirical probability distribution fitted from the real patients.

Similarly, to generate patient-specific parameters we first studied the correlation between parameters of real patients (see Table 1). When strong dependencies among parameters appeared (Spearman correlation coefficient $r_s > 0.75$), they were translated to the generation of virtual patients to have parameters generated to be as faithful as possible. We used linear regression to model the relationships between correlated parameters and also explored multivariate regressions, but the latter were disregarded in favor of the more simple linear ones. We used Cholesky decomposition to generate random values of the parameters following the linear model derived from the real data preserving the correlations inferred from patients.

The number of TMZ cycles administered to the virtual patients was first drawn from a discrete normal distribution with a mean of 19 cycles and a variance of 7, excluding values lower than 6 and higher than 34 to avoid unrealistic values (S2B Fig). This was done to reproduce the same behavior of the real patients and avoid generating artificial comparisons. Later on, in order to give rise to a protocol proposal, we carried out trials in which the number of doses received by every patient was fixed and equal for every patient.

Inferring a precise statistical distribution for the fatal volume is complex due to the lack of clinical data. Previous studies used the value of 280 cm$^3$ [18]. Building on that information, we drew the fatal volume from a normal distribution with a mean of 280 cm$^3$ and a variance of 20 cm$^3$ to add variability between virtual patients and account for the fact that tumor location may impact the fatal volume (S2C Fig).

## In silico clinical trials

We created in silico clinical trials in which the individuals were random virtual patients created following the method described above. Different protocols of application of TMZ were tested on large cohorts to identify a scheme that improves the OS of patients.

Each in silico clinical trial considered 100 patients per arm. In the first arm, the virtual patients were treated by the traditional C28 protocol, while in the second arm, they received an ID protocol consisting of individual doses spaced by rest days in periods of discrete weeks. Each in silico clinical trial was launched with a different random seed. The simulation of each virtual patient was started 30 days prior to the treatment initiation. The end point corresponds to a death event, therefore the simulation stops when the tumor reaches the critical size.

To compare the difference in survival between different arms in the clinical trial, we used the Kaplan-Meier estimator. Significance of the difference between survival curves was

evaluated with the log-rank test. Kaplan-Meier survival curves and log-rank test were realized respectively with the function `KaplanMeierFitter` and `logrank_test` from the `lifelines` Python package version 0.27.1 [41].

# Results

## ODE model describes the evolution of patients receiving TMZ treatment

We evaluated the capability of our model to fit the longitudinal tumor volumes of the seven LGG patients who showed acquired TMZ resistance. Times from diagnosis and volumes can be found in supporting information S1 File. Fits for each patient are depicted in Fig 3A–3G, with the best fit parameter values presented in Table 1.

For each patient, we simulated the same chemotherapy treatment that the patient had initially received, namely a classic cyclic protocol C28 consisting of a single oral dose of TMZ (consider as equal to $E(t_{T,i}) = 1$ in the model) per day for 5 consecutive days, followed by 23 days of rest, for a given number of cycles depending on the patient, as it is used in clinical practice [42].

Three patients had additionally received radiotherapy: patients 1, 6 and 7. Patient 1 (Fig 3A) received radiotherapy after TMZ treatment had finished, thus not interfering with the fit. The patient developed resistance to the TMZ treatment as evidenced by the lack of response to the second administration of chemotherapy. The late application of the radiotherapy did not interfere with our results on chemotherapy. Patient 6 (Fig 3F) received radiation therapy prior to TMZ administration and right after diagnosis. It is difficult to quantify the precise effect of radiotherapy, but its application being independent of chemotherapy, and the actual growth in volume seen after its application for this specific patient, encouraged us to keep these data in our cohort and focus only on the later chemotherapy treatment and regrowth with evident resistance development. Patient 7 (Fig 3G) underwent a Stupp protocol consisting of concomitant radiotherapy (2 Gy per fraction) and TMZ (considered as equal to $E(t_{T, i}) = 0.5$ in the model) for 6 weeks, every day of the week, except on weekends, followed by a classical cyclic TMZ chemotherapy [43]. In order to follow the focus on TMZ administration we did not consider within our modeling framework the—potentially important—effect of radiotherapy. However, the model was able to fit well the last part of tumor evolution, where the effect of radiotherapy is more likely to have vanished and the resistance acquisition to TMZ is more relevant. For all other patients, fits were close to real data.

In addition to the data points from the patients data and the fit of the total volume provided by our model, we also represent in Fig 3 the behavior of the individual compartments considered in the model. After each dose of TMZ, part of the sensitive population becomes damaged and another part turns to the persistent state. Whereas the drug effect causes the delayed death in the damaged cell compartment, it is also responsible for the transformation of persistent cells into fully resistant when an additional dose is given. The persistent cells that do not transform become sensitive again following an exponential decay.

Obtained parameter values represented in Table 1 allowed to mimic the observed macroscopic tumor growth and the emergence of TMZ resistance as described in Ref. [12]. Importantly, our mathematical model was robust and fitted the different dynamics observed with values of the parameter within similar ranges for all patients.

## Alternative protocols improved OS in virtual in silico twins

Using the set of parameters that best described the evolution of the patients, we created virtual twins and applied different TMZ in silico protocols to evaluate their effect on the tumor shrinkage. We qualitatively compared the effect of the new protocols with the classic C28

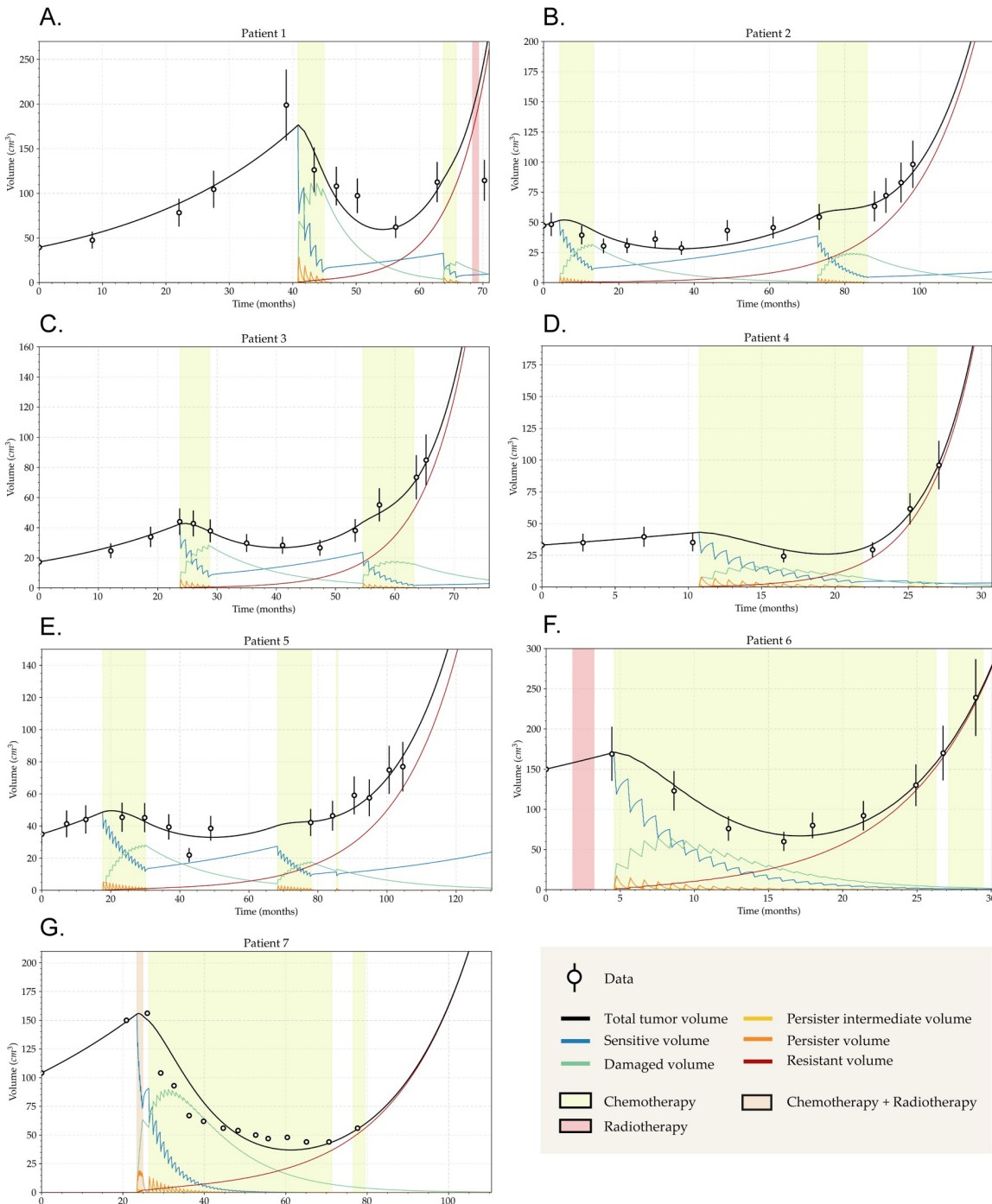

**Fig 3. The model describes the longitudinal evolution of the tumor volume.** Fits of all patients longitudinal growth to the model Eqs (1–6) obtained by minimizing the root mean square error between data and the model. These patients mainly received a cyclic TMZ treatment (yellow background), some of them also received radiotherapy (red background) and one of them received both (light brown). (A-F) Volumetric data were acquired by ellipsoidal approximation from the MRI images, error bars represent 18% of error on tumor volume. (G) Volumetric data were acquired by precise semi-automatic segmentation and therefore no error bars were included. Each volume associated to a cellular population of the model was represented by its own curve. Each dose induced drives sensitive cells into either the damaged or the persister cell compartments. Without a supplementary dose, persister cells go back to the sensitive phenotype. Whereas, with a supplementary dose, a small proportion of persistent cells become resistant. As the simulation progresses, resistant cells become the majority and the tumor no longer responds to TMZ.

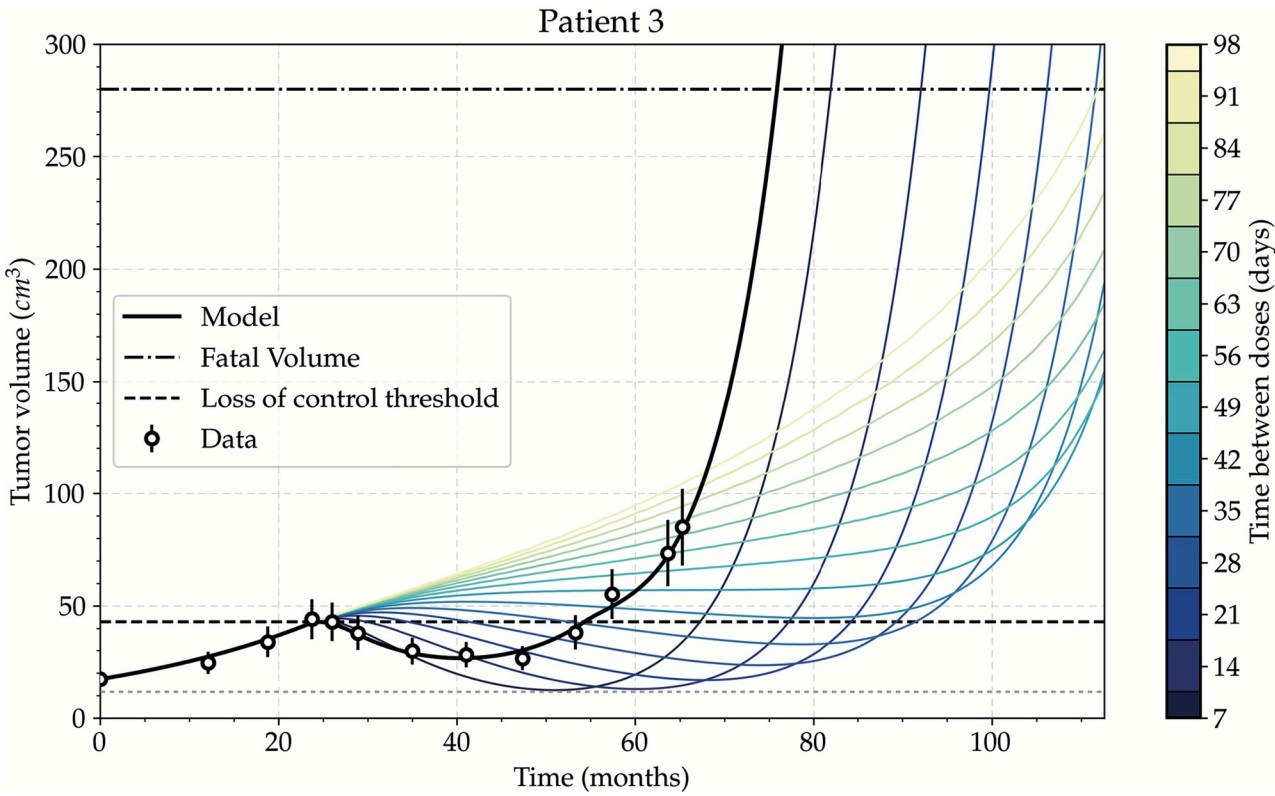

**Fig 4. Effect of different time intervals between individual doses on tumor growth.** Simulation of different experimental protocols consisting of spacing each single dose by a given number of days, from 7 to 98. This patient received 80 doses of TMZ according to the classic cyclic protocol (black line). The longer the interval between doses, the longer the time to reach fatal volume (OS). However, tumor control is lost from 42 days between each dose. Error bars represent 18% of error. Inferior horizontal line shows minimum attained volume.

protocol. Fig 4 shows the simulation of virtual protocols on patient 3, doses applied individually with dosing intervals ranging from 7 to 98 days. In all the cases, 80 doses were given, what corresponds to the number of doses the patient received during the C28 protocol.

These simulations show that longer rest periods between doses lead to improvements in OS. However, as the interval between doses gets larger, the control over tumor growth is eventually lost. In particular, for patient 3 the treatment does not induce a tumor shrinkage when the dose spacing reaches 42 days; at that point tumor growth is no longer controlled. Higher tumor volumes are correlated with a bad prognosis [44], for instance because of an increased of intracranial pressure [5] or higher risk of malignant transformation [45]. Also, in clinical practice, tumor growth during treatment leads to its discontinuation. Consequently, with dosing intervals of 42 days the protocol would not be suitable anymore.

Similar dynamics were observed for other patients (see S3 Fig), nevertheless, the loss of control thresholds for other patients, like patient 1 (S3A Fig) and patient 7 (S3G Fig), are higher. Between all the patients, we found a mean rest period causing loss of control of 55 days, with a standard deviation of 17.7 days. The threshold therefore depends on the specific patient. Since the goal was to obtain a general protocol improving the outcome for all patients, any scheme for which there is an increase in tumor volume for any single patient should be neglected. As a consequence, we did not consider any ID protocol that spaced doses over 42 days.

Our results thus showed that spacing out each single dose between 7 and 42 days increases both OS and tumor shrinkage for all selected patients of our dataset.

## Generated cohorts of virtual patients extend the information from real data

Pairwise correlation matrix of real patient parameters are shown in S4 Fig. Four pairs of parameters were highly correlated ($r_s > 0.75$): $\rho_1$ and $\psi$ ($r_s = 0.82$), $\tau$ and $\rho_2$ ($r_s = 0.79$), $\alpha_1$ and $\psi$ ($r_s = 0.79$), and $\alpha_1$ and $\alpha_2$ ($r_s = 0.95$). We fitted the relationships between these variables by linear models as seen in S4B1 Fig and used these to generate correlated parameters for the virtual patients, which followed the relationships shown in S4B2 Fig.

Cohorts of virtual patients were created as described in the 'Methods' section by using random parameters in the ranges of the parameters inferred from the real data and respecting the observed correlations. The correlations of the parameters between virtual patients are depicted in S4B2 Fig, where the similarity between real and virtual cohorts is apparent.

## In silico clinical trials show that protocols with individually spaced doses have an OS benefit

The results of the previous section show the effectiveness of spacing out individual doses. To obtain stronger results, we implemented an in silico clinical study in which we set up different experiments based on cohorts of virtual patients.

Using cohorts of 100 virtual patients per arm, we investigated ID protocols with the goal of finding schemes with better OS than the standard C28. We studied ID schemes with a dosing interval between doses of 7, 14, 21, 28, 35, and 42 days. Kaplan-Meier survival curves of representative in silico clinical trials for the different ID protocols tested are shown in Fig 5. All ID protocols showed a better survival than the C28 standard scheme.

To further substantiate our results, for each ID protocol, we simulated a total of 20 clinical trials with 100 virtual patients per arm. To measure the survival gain, we computed the difference at the median between the survival curves of each arm. The difference of median survival between the ID schemes for different rest periods with respect to the C28 are shown in Fig 6A. Greater dosing intervals between doses lead to survival gains in the median between 5 months for the ID7 to 27 months for ID42.

In simulations of ID treatments on virtual clones of real patients, the control of tumor growth is lost when a certain number of weeks between each dose is exceeded. Therefore, there is a trade-off between tumor shrinkage and overall survival. We performed a second experiment consisting in simulating 500 patients under ID7 to ID42 protocols. For each patient, we computed the difference between the post-treatment minimum tumor volume and the volume at the beginning of the TMZ treatment. Results shown in Fig 6B demonstrate that the larger the spacing between doses, the smaller the loss of volume.

In order to show that the conclusion drawn from the clinical trials is independent of the fixed parameter $\beta$ ruling the transition of persister cells back to the sensitive state, we performed 4 new in silico clinical trials using four different values of $\beta$ and using the comparison between the ID21 protocol and the C28 protocol as a reference (S5 Fig). The results showed that the previous conclusions hold for all the tested values of $\beta$, in the range $[2 \times 10^{-2}, 1]$ day$^{-1}$. Moreover, we tested the dependency of the results on the shape of the switching function between $V_{PI}$ and $V_P$, ruled by a single parameter in the sigmoid curve. We ran two additional in silico clinical trials of the ID21 protocol versus the C28 with two different shapes of the said function (S6 Fig) and found no difference with the previous results.

To guarantee the safety of the proposed protocols, we established a new requirement on the control of tumor volume: No virtual patient enrolled in the trial can experience loss of volume control. Therefore, ID protocols in which at least one patient experienced loss of control were neglected. This happened for protocols ID42 and ID35, for whose 20 respective clinical trials,

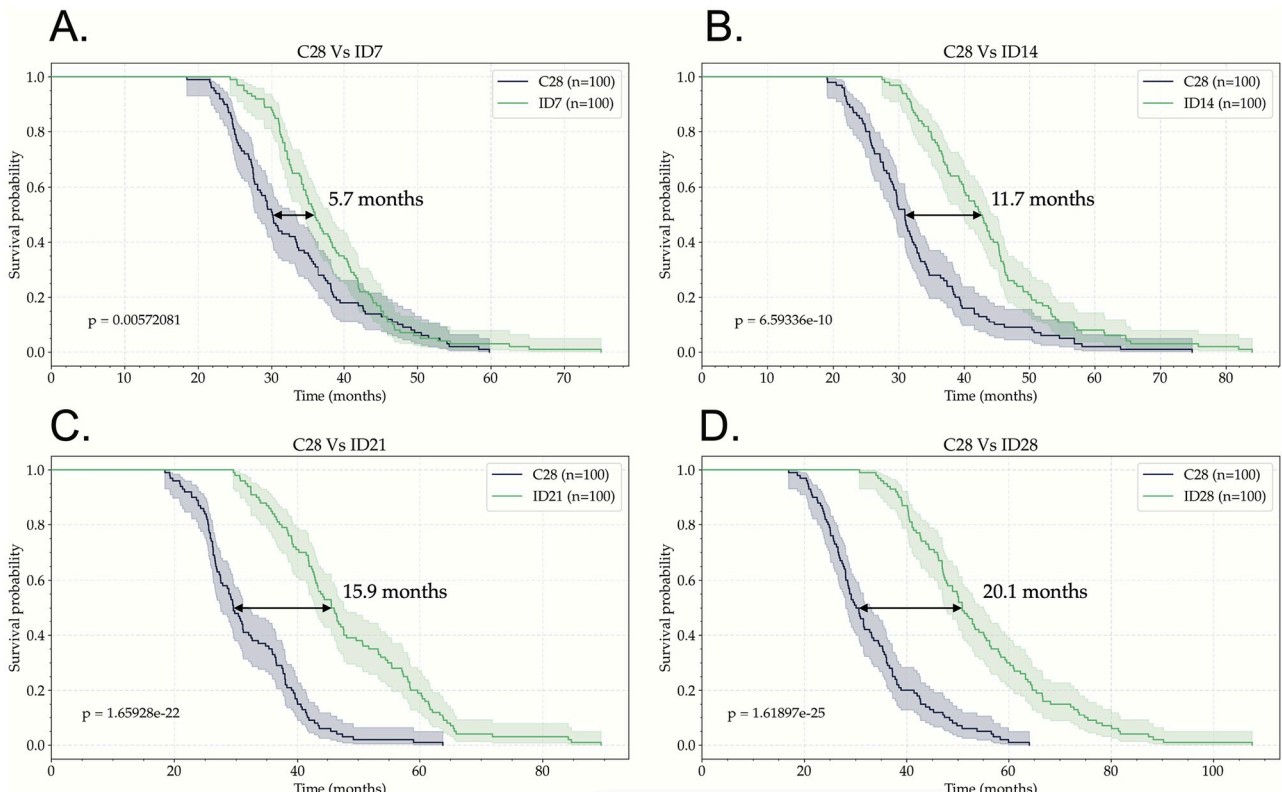

**Fig 5. ID protocols have a better survival curve than the classic C28 protocol.** Kaplan-Meier survival curves of four virtual clinical trials which consisted of comparing C28 protocol with the administration of individual TMZ doses spaced by: (A) 7 days (ID7), (B) 14 days (ID14), (C) 21 days (ID21), and (D) 28 days (ID28). Each arm consists of 100 virtual patients. TMZ treatment was initiated 30 days after the start of the simulation for all patients. p-values were calculated using the log-rank test.

there was at least one patient for which the volume could not be stabilized by the treatment. Therefore, these two protocols were neglected.

Therefore, even though large dosing intervals might be beneficial for some cases, intermediate spacings like ID14 and ID21 provide a better compromise between OS gain and reduction of

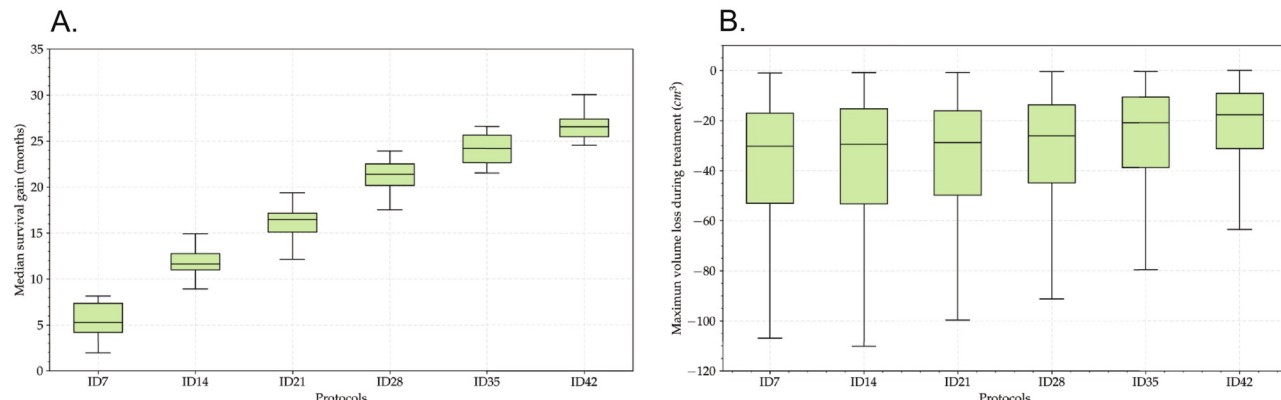

**Fig 6. Compromise between OS and tumor shrinkage.** (A) Boxplot showing the survival gain (calculated at median of Kaplan-Meier curves) of different ID protocols versus the C28 one. For each protocol, 20 clinical trials with 100 patients per arm were simulated. (B) Boxplot showing the maximum volume loss when the time between individual doses is spaced from 7 to 42 days. For each protocol 500 patients were simulated.

tumor volume. These two protocols show safety in controlling tumor volume while at the same time provide significant improvements in terms of overall survival. Additionally, they have the advantage of their easy clinical implementation, since they propose giving a dose regularly every two weeks, or every three weeks, respectively. As a consequence, we propose using ID14 or ID21 protocols for TMZ administration in LGG patients as a means of safely extending overall survival.

### Individual dose protocols show superiority in virtual clinical trial with fixed number of cycles

In the previous analysis we intended to get as close as possible to the real situation represented by our patient's dataset. Since every patient received a different number of cycles, we reproduced the same behavior in the in silico patients by generating virtual cases which received a random number of total doses (within the data ranges). This number of doses was therefore different for each patient.

However, demonstrating the efficacy of a proposal for a new standard by means of a clinical trial would imply delivering a pre-fixed and equal number of cycles to every patient. Therefore, we ran new clinical trials that fulfilled this criterion. For the ID14 and ID21 protocols, we set two different clinical trials to compare them to the C28 standard. The first delivered a total amount of 12 cycles to each of the virtual patients enrolled, while the second was based on a 24 cycles rationale.

The Kaplan-Meier curves of these trials (S7 Fig) show that the ID14 protocol entails a gain of around a year in median overall survival, while the ID21 is able to provide a median survival gain of 13.8 months for the study with 12 cycles and 15.2 months for the study with 24 cycles.

### Clinical trials should enroll at least 40 patients per arm to demonstrate the superiority of ID schemes

The computational results shown here need a validation on patients to confirm the results. It would be desirable to minimize the number of patients enrolling a clinical trial. Therefore, we calculated here the minimum number of patients required to prove the benefit of the proposed ID14, ID21 and treatment schemes in a real clinical trial.

For each of the protocols ID14 and ID21, we organized 20 independent clinical trials with 10 patients per arm. The trials were repeated by gradually increasing the number by 10 patients per arm each time until a total of 100 patients per arm. This gives an outlook of the potential results from clinical trials. The goal was identifying the minimum number of patients for which the trial gave a significant difference in survival between the virtual patients in the groups. For the results of each of the clinical trial we evaluated their significance by the p-values of the log-rank test. Results are shown in Fig 7.

For ID14 protocol, 30 patients per arm produced a significant trial 95% of the times. With 40 patients per arm all trials were significant. In the case of ID21 protocol, 90% of the trials were significant with 20 patients per arm, whereas with 30 or more patients per arm all were significant. As a consequence, the minimum number of patients per arm to be completely sure of proving the superiority of ID14 and ID21 protocols are, respectively, 40 and 30. Notice that the enrollment of 80 patients with the desired characteristics might involve the initial consideration of a higher number of patients of which many might be rejected.

### Adaptive variations of ID protocols can be used to limit toxicity

The adaptation of the treatment schedule in a per-patient basis in order to avoid the emergence of resistant phenotypes and to minimize side effects has been studied theoretically

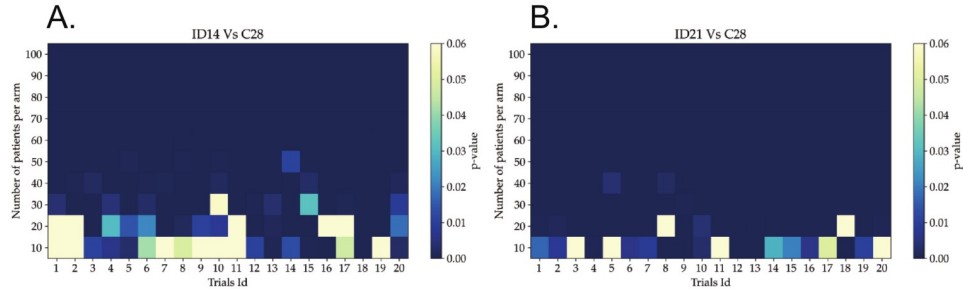

**Fig 7. Required number of patients to prove the benefits of ID protocols.** p-value matrices of (A) ID14 protocol versus C28 protocol and (B) ID21 protocol versus C28 protocol. Each column represents an independent clinical trial in which the number of patients was progressively increased. The probability of obtaining a significant trial applying the ID14 protocol is 100% with 40 or more patients per arm. However, for the ID21 protocol, at least 30 patients per arm are needed.

extensively in recent years [46]. The concept has proven useful in some sets of prostate cancer patients [47] and is under testing for other cancer histologies [48]. This so-called adaptive therapy is based on controlling the disease burden below a certain limit as assessed by a biomarker. Once the disease burden is below the target value for the biomarker treatment is reduced or even removed. When the disease markers are out of range treatment is restarted or adapted dynamically. The simplest way of using this idea is implementing on/off treatment cycles depending on the biomarker values.

We applied adaptive therapy to virtual patients simulated through the model (1)–(6). The time between doses was set depending on each protocol (ID14, ID21) and the MRI screening time was set to 90 days, as it is typically done in the clinical practice for LGG. The volume readings obtained in the screening was used to implement therapeutic decisions in silico. If the tumor volume was below a certain threshold, all doses until the next screening were spared. No treatment was applied until, in subsequent screenings, the tumor volume was found to be over the threshold chosen. In that case, the dosage schedule was resumed until the next screening. We show an example of the application of this therapy in Fig 8A. In the simulation shown there, we compared the total tumor volume using the ID21 protocol and the one of the adaptive protocol. The respective cell subpopulations obtained when using the adaptive protocol are depicted there. We also tested whether the screening time has a big influence on this specific case and found a low sensitivity to that parameter due to the small length relative to the overall period where the therapy is activated/deactivated (S8 Fig).

To check if adaptive therapy resulted in an improvement of survival with respect to the continuous ID14 and ID21 protocols, we performed several virtual clinical trials with adaptive therapy for different dosages and volumetric thresholds ranging from 30% to 80% of the initial tumor volume (S9 and S10 Figs). The results of these virtual clinical trials showed that there is no significant difference in terms of survival between using the ID protocols or their adaptive variation (8(B)–8(C)). Therefore, skipping dosages by means of adaptive therapy can serve as a way to limit toxicity and side effects of the treatment, while preserving the gain in survival given by the ID protocols.

## Discussion

In this work we studied the response of LGG to TMZ via a mathematical model of ODEs that includes the acquisition of resistance to treatment through the recently discovered path of persister cells [10, 14]. Our approach follows a trajectory of previous works based on similar

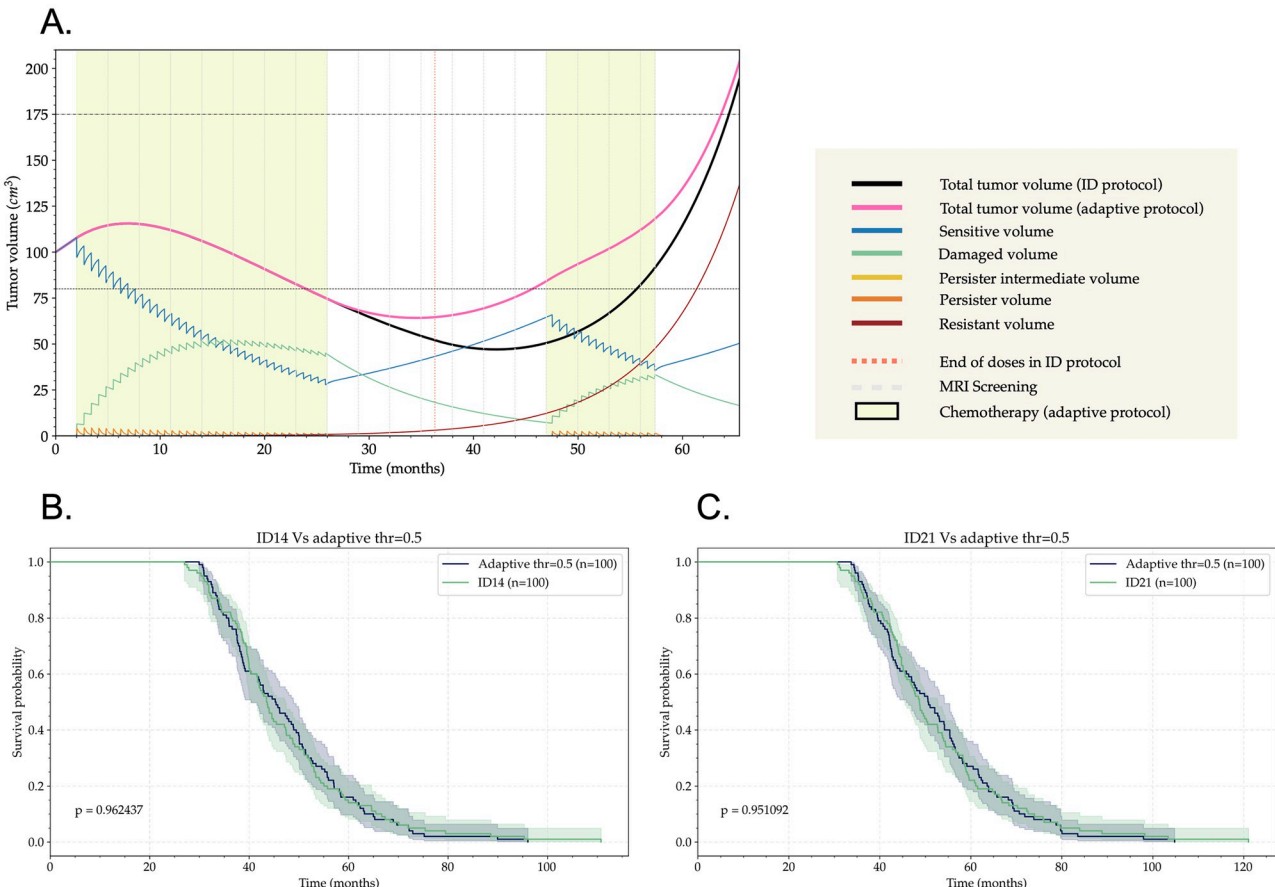

**Fig 8. Adaptive therapy joint with ID protocols may ameliorate the toxicity of the treatment while maintaining the survival gain.** (A) Time evolution of tumor volume for a virtual glioma patient undergoing TMZ under two different protocols. The black line corresponds to an ID21 protocol where 50 doses are equally spaced in time. The time at the end of the dose is indicated with a vertical dotted orange line. The pink line corresponds to the tumor volume with that same ID protocol but applying an adaptive variation. In this case, the patient is screened every 90 days (gray dashed lines). During the next interval between screenings, the doses are only applied if the tumor volume is higher than a certain proportion (taken as the 80% here) of the initial volume. The time periods when the doses are applied are indicated by a yellow background. (B) Results of the virtual clinical trial were generated by comparing the ID14 protocol with its adaptive variation using a threshold of 50% of the initial tumor volume. (C) Results of the virtual clinical trial were generated by comparing the ID21 protocol with its adaptive variation using a threshold of 50% of the initial tumor volume.

models for the description of LGG growth and response to treatment, as ODEs are the right tool for the description of longitudinal time dynamics and the interaction of populations or species. Ribba et al. proposed a model for tumor diameter growth in LGG that was based on three evolving populations affected by the chemo and radiotherapy [19]. Their model was able to describe size evolution, but did not consider the acquisition of resistance by cells exposed to the drugs. Later on, Mazzocco et al. considered that dimension [49], while also including the genetic characteristics of the patient [50]. On a different line, Badoual et al. focused on the role of edema reduction to understand the delayed response of tumor volume to radiotherapy [51]. Ollier et al. studied the resistance to treatment induced by the therapy and tested their hypothesis on a big cohort of 121 LGG patients [52]. Studying the acquisition of resistance in gliomas, Trobia et al. introduced an ODE model which also accounted for the glia-neuron interaction [53]. Pérez-García et al. developed a new ODE model which successfully identified how tumors showing a faster response to therapy are actually more aggressive [32], work that was continued by Bogdańska et al., who provided a paradigm for probing tumor with TMZ [17,

54]. Later on, Pérez-García et al. proposed modified TMZ application protocols with the goal of improving survival [18]. In particular, this last work did not include any feature regarding the evolutionary dynamics of resistance acquisition; in that sense, it was based on a simpler model which was unable to represent the details of the underlying biology. In contrast to the mentioned works, our approach here includes the recently found biological principles of persister cells as a path to resistance, as well as data from patients that had developed resistance during the treatment with TMZ.

Resistance is a major problem in TMZ chemotherapy [55], and an understanding of the mechanisms behind its acquisition may allow the adaptation of TMZ protocols to delay its emergence. Here, we developed a simple mathematical model based on recent knowledge about human persistent cancer cells (or persisters) [7, 12, 13] to describe the macroscopic evolution of LGG and the acquisition of resistance to TMZ. The process of resistance to TMZ in low-grade gliomas has been studied by mathematical models before [52, 56] but, to our knowledge, no work has incorporated the intermediate persistent state during that process. In their work, Ollier et al. [52] investigated the mechanism that best explains the origin of the resistance to TMZ: acquired (due to epigenetic changes or genetic mutations on cells that are originally sensitive to treatment), or primary (due to the natural ability of the tumor cells to become resistant). However, not all the included patients showed evidence of developing TMZ resistance. Tumors are usually spatially heterogeneous and it cannot be excluded that primary resistance may coexist with acquired resistance, but it seems that it is the latter that is behind the emergence of TMZ resistance through TMZ induced mutations and hypermutations [4, 57–59]. Here we considered only this last hypothesis, which is in agreement with the research considering persisters.

The development of TMZ resistance involves a complex interplay of numerous molecular mechanisms. These include epigenetic mechanisms, such as the methylation of the CpG island at the MGMT gene promoter, and also genetic mechanisms, such as the loss of function in mismatch repair (MMR) genes, the overexpression of ATP-binding cassette (ABC) transporters [60], or the activation of ataxia telangiectasia mutated kinase (ATM) [61, 62]. Other effects, like gene regulators of autophagy as ATG9B, also have an effect in acquired TMZ resistance [63], as does the vesicle-associated membrane protein 8 (VAMP8), which significantly increases cell proliferation and TMZ-resistance [64], in agreement with our results regarding $\rho_2 > \rho_1$. Moreover, MMR defects due to TMZ exposure have been found to lead to hypermutation in recurrent gliomas [65], which show a transformed genetic landscape as a consequence of the treatment. Therefore, as a consequence of the multiple mechanisms involved in the acquisition of TMZ resistance, and the fixed nature that many of them share, that phenomenon must be irreversible. Taking together all these evidences, we chose to model the resistant compartment $V_R$ as an irreversible state.

Our model was validated through imaging data of LGG patients which had shown acquisition of resistance to TMZ. Semi-automatic segmentation on high quality FLAIR MRI images was used to obtain accurate longitudinal volumetric measurements. Indeed, accurate data acquisition is necessary to know exactly when the growth dynamics of a tumor changes, what is required to study the onset of resistance. The results from fits of volumetric growth to our model showed a strong dependency among several parameters. These correlations had to be taken into consideration for the generation of cohorts of virtual LGG patients. The correlation between $\alpha_1$ and $\alpha_2$ is artificially induced by the constraint of $\alpha_2$ according to the value of $\alpha_1$. Other relevant correlations appeared between $\rho_1$ and $\psi$, and $\rho_2$ and $\tau$. This means that the larger the growth rate of sensitive cells, the more of them transit to the damaged compartment and the faster the damaged cells die, the faster the regrowth rate is. Thus, the mechanisms associated with tumor response to therapy appear to be positively correlated with the sensitive

population growth rate. This is in agreement with the fact that TMZ damages mainly dividing cells [66]. Regarding the other parameter correlation, Pallud et al. [24] observed a worse prognosis for low-grade glioma patients displaying faster tumor shrinkage after therapy. More recently, Plaszczynski et al. reported a positive correlation between the proliferation rate and the death term [67] what agrees with the correlations between $\rho_2$ and $\tau$ found here. An interesting feature obtained in our parameter fitting was that the value of the parameter $\beta$, related to the time that persistent cells take to return to a sensitive state, is similar for all the patients. This points to a characteristic biological time of the process of around 7 days. This value has important consequences, as any treatment scheme intending to maximize the return of persisters to the sensitive state should allow at least that amount of time between doses. As our results confirm, protocols in which the TMZ doses are spaced two or three weeks improve the survival of the patients by reducing the generation of resistance via that mechanism. Ideally, it would be good to have a model describing the same biology with fewer parameters, especially taking into account that the model has six free parameters and the patient with fewer data points has eight follow-ups. However, such a model would not be able to describe equally well the relevant underlying biology. Also, other patients have up to 15 data points and the model have been shown to fit them reasonably well, what supports the validity of our method.

Regarding the growth models considered in our model for the resistant population $V_R$, we have chosen a proportional term that yields an exponential unbounded growth. Here, the proliferation rate $\rho_2$ is a net proliferation that accounts both for the intrinsic proliferation of the population as well as the dead of resistant cells. As the balance between born cells and dying cells is positive, the net proliferation $\rho_2$ yields a positive growth term. Notice that growth limitations could have been included in this equation, being the most traditional ways the addition of a logistic term $(1 - V/K)$ or a Gompertz term $(\log(K/V))$. In both cases, these terms set a limit to the overall size that the tumor can reach, and have the additional cost of introducing a new parameter. Particularly, for the case of the logistic term, the results in the low volume range remain unchanged as the term vanishes for low $V/K$. This poses the problem of identifying the actual value of the carrying capacity $K$, as any arbitrarily big value of the parameter would yield results identical to those of a model that does not include this term; therefore, our model could be equivalently written in that way with unchanged results. Moreover, the literature analyzing the longitudinal growth of untreated LGG points to a continuously accelerating growth [68, 69]. Thus, in agreement with the previous facts, we have disregarded the kind of growth limitations discussed.

We used here a limited number of patients with the desired characteristics. We only included biopsied patients in our study, in order to avoid the radical changes on volume in patients that undergo partial or full resections. Besides, we only considered TMZ treatment and neglected the role of radiotherapy, which is an important part of the clinical practice [43] and could have a relevant effect in the evolution of our patients. The mathematical modeling of combined chemotherapy and radiotherapy is still poorly understood and it is difficult to estimate the individual impact of the two treatments when they occur at the same time as each patient responds differently to these treatments [28, 70]. As future work, it would be interesting to explore further the combined effect of chemotherapy and radiotherapy and propose optimum combined protocols. Further analyses on additional patient datasets are required to investigate whether tumor location, molecular type and combination of treatments may impact treatment response or resistance mechanisms. Moreover, a higher number of patients in the study would entail a better representation of the disease in our models.

When the model was validated through patient data, we studied how modified protocols of treatment administration could improve the OS of patients by delaying the emergence of TMZ resistance. We chose to study schemes in which the doses were given individually with variable

periods of rest between them. These periods were designed in multiples of weeks in order to keep the protocol simple to implement. We simulated these schemes computationally on virtual copies of the patients and on other synthetic patients to perform in silico clinical trials. Our simulations showed that spacing individual doses in periods of 14 or 21 days, i.e. beyond the persister characteristic lifetime ($\beta$), yield a good compromise between increasing OS and controlling tumor growth. A small dependency of the survival gain with respect to the parameter $\beta$ can be seen (S5 Fig). For the sake of the sensitivity analysis, we used a wide range of parameters that might not be found in real patients. However, if that was not the case, that dependency of the survival gain on the parameter $\beta$ would mean that the parameter could be analyzed from anatomical pathology samples to use it as a biomarker discriminating patients that would benefit more from ID protocols. In the study by Segura et al. [13] different time intervals between individual doses of TMZ were tested in an in vivo model of slow-growing glioblastoma. Compared to other protocols with lower dosing interval between individual doses, 14-days periods showed a significant decrease in resistance markers and, importantly, an increase in overall survival. They hypothesized that these results could be true for low-grade gliomas, which are also slow growing tumors. Our results support their hypothesis using data obtained directly from LGG patients. Also in comparison to their study our model was substantially simpler. Although some biological facts were similar, our more efficient computational approach allowed us to scan larger regions of the parameter space and perform a more accurate parameter fitting.

Several types of adaptive therapy have been proposed potentially providing survival improvements in different scenarios [47, 71, 72]. The simplest adaptive approach that can be implemented is the treatment suspension while the value of a biomarker of the disease burden is below a preset threshold. This may have a positive effect of reducing treatment-induced resistance. LGG ID protocols would seem a good candidate for adaptive therapy, as they have long times with continuous follow-up and are incurable by nature. The tumor volume as seen in MRI could serve as a marker for the decision of cessation or resumption of the treatment. We considered the application of adaptive therapy in combination with the previously discussed ID14 and ID21 protocols to check whether they could provide clinical benefits. To do so, we simulated MRI screenings to check if the patients' tumor volume was under a predefined threshold. At imaging times where volumes are obtained, typically every three months, dose administration would be stopped and later resumed whenever the volume exceeded such threshold. Notice this kind of adaptive therapy would reduce the number of doses received by the patient, and thus the side effects caused by the exposure to TMZ. The results of the virtual clinical trials showed that this kind of therapeutic scheme did not reduce the survival benefit of the ID protocols and can therefore be used to limit the toxicity of the treatment. Such simulations show that, even when fatal tumor volumes were reached at almost the same time as in the unaltered ID protocols, several dosages, and therefore toxicity, can be spared. A key idea behind adaptive therapy is that there is a fitness cost to resistance, so that the therapy-sensitive population can outcompete the therapy-resistant population in the absence of treatment. Without this relationship, removing therapy does not control the resistant population due to the model assumption that TMZ resistant cells gain fitness. Therefore, even though the adaptive variation may serve to limit toxicity, we do not see survival benefits of it within our modeling framework. In the future, it would be interesting to consider whether the emergence of TMZ-resistant cells in LGG entails any fitness cost not described here that might justify a regression of this population in the absence of treatment. Such an approach might discover improvements in the adaptive therapy in terms of survival in addition to the limitation of toxicity shown here.

Other in silico studies have suggested that spacing TMZ cycles or radiotherapy fractions could increase overall survival [18, 73]. In Ref. [18] the authors suggest an alternative protocol in which the cycles of TMZ are given in daily shots as usual, but are spaced out in time. They also include an induction period consisting of five induction cycles. In the view of our current modeling approach, supported by recent advances in the understanding of resistance acquisition, just as an approach would be detrimental for the patient in the sense that the initial induction period would lead to a fast generation of resistant cells. Several evidences go in the direction of our results suggesting a less frequent administration of doses. However, 21-day intervals between individual doses of TMZ have never been tested in vivo and are still just a theoretical proposal. These in silico results pave the way for potential future trials that may result in improvements in the treatment of LGG patients. Our study suggests specific variations with an easy clinical implementation that could raise the efficacy of an approved drug like TMZ. The clinical implication of an individual dose-spacing protocol for TMZ is twofold. First, it delays the emergence of resistance by allowing time for persistent cells to return to a sensitive state. For patients, this translates into increased overall survival, while maintaining control over tumor growth. Second, reducing the frequency of TMZ dosing would reduce treatment toxicity. The quality of life of patients would therefore be improved.

Our model also opens the way to personalized medicine of LGG patients. We identified a general protocol that would be beneficial to all patients. Nevertheless, we can go further by implementing a method able to find the optimal protocol according to patient individual parameters. This would allow to take into account the heterogeneity of behaviors observed in these tumors. Since low-grade gliomas are slow-growing tumors, there are often non-treatment follow-up periods during which tumor volume measurements on MRI scans are available. Such measurements may allow model parameters to be well characterized and then used to find personalized optimal protocols.

## Conclusion

We developed a mathematical model of LGG growth and response to TMZ. We incorporated persister cells, as path to chemotherapy resistance. The model was able to describe the volumetric evolution observed in routine MRIs of patients treated with TMZ and with evidences of having acquired resistance. The values of the parameters that best described the evolution of each patient were used to computationally test protocols of drug administration in which doses are given individually with resting periods multiple of weeks between them. According to our in silico results, the protocols with individual doses spaced by dosing intervals of 14 or 21 days would produce, in terms of OS and tumor reduction, the best treatment outcomes. Adaptive approaches may lead to even lower toxicity but did not improve survival in silico.

## Supporting information

**S1 Fig. Segmentation and ellipsoidal approximation.** Axial (A), coronal (B) and sagittal (C) slices of FLAIR 3D MRI study of patient 7 at the time of diagnosis. The blue mask indicates the area delineated semi-automatically using a gray-level threshold. Above it, the white arrows indicate the measure of the diameters used for the ellipsoidal approximation of tumor volume. (D) 3D reconstruction of the segmented tumor.
(PDF)

**S2 Fig. Statistical distribution used to construct the virtual patients.** Statistical distribution of initial tumor volume, number of TMZ doses and fatal volume used to construct the virtual

patients.
(PDF)

**S3 Fig. Effect of different time interval between individual doses on tumor growth.** Simulation of different experimental protocols consisting of spacing each single dose by a given number of days, from 7 to 98 (gradient lines). The longer the interval between doses, the longer the OS. However, tumor control is lost after a certain number of days between each dose, depending on each patient. Error bar represent 20% of error.
(PDF)

**S4 Fig. The strongest pairwise linear correlations are preserved in the virtual patients.** (A1) Correlation matrix between real patients parameters. The white stars represent significant correlation (n = 7). (A2) Correlation matrix between virtual patients parameters (n = 200). (B1) Linear regression model of correlated parameters of real patients (n = 7). (B2) Linear regression model of correlated parameters of virtual patients. Three of the four significant correlations, the most biologically relevant, are preserved in virtual patients. Those correlations appear to be linear relations.
(EPS)

**S5 Fig. Effect of the parameter $\beta$ on the results of the virtual clinical trial.** Results of clinical trials of the ID21 protocol against the C28 protocol with four different values of the fixed parameter $\beta$. Kaplan-Meier curves are shown with the results in terms of survival of these clinical trials. (A) $\beta = 2 \times 10^{-2}$ day$^{-1}$. (B) $\beta = 5 \times 10^{-2}$ day$^{-1}$. (C) $\beta = 5 \times 10^{-1}$ day$^{-1}$. (D) $\beta = 1$ day$^{-1}$.
(PDF)

**S6 Fig. Effect of the switching function $f(E)$ on the results of the virtual clinical trial.** Results of clinical trials of the ID21 protocol against the C28 protocol with two different values of the fixed parameter that modifies the shape of the switching function between $V_{PI}$ and $V_P$. Kaplan-Meier curves are shown with the results in terms of survival of these clinical trials. (A) $f(E) = 1/2(1 - \tanh((E - 0.05)/0.05))$. (B) $f(E) = 1/2(1 - \tanh((E - 0.1)/0.1))$.
(PDF)

**S7 Fig. Results of clinical trials of the ID protocols with fixed number of cycles.** Kaplan-Meier curves showing the distribution of overall survival in virtual LGG patients undergoing TMZ treatment with ID14 and ID21 protocols (individual doses given every 14 or 21 days) against the classical C28 regime. Importantly, the number of drug cycles pre-set for all the virtual patients enrolled in the trial is the same. (A) Results for ID14 with 12 cycles. (B) Results for ID14 with 24 cycles. (C) Results for ID21 with 12 cycles. (D) Results for ID21 with 24 cycles.
(PDF)

**S8 Fig. Additional simulations of adaptive therapy for an individual patient with different screening times.** Time evolution of tumor volume for a virtual glioma patient undergoing TMZ under two different protocols. The black line corresponds to an ID21 protocol where 50 doses are equally spaced in time. The time at the end of the dose is indicated with a vertical dotted orange line. The pink line corresponds to the tumor volume with that same ID protocol but applying an adaptive variation. (A) The patient is screened every 30 days (gray dashed lines). (B) The patient is screened every 180 days (gray dashed lines). In both cases the doses are only applied during the next interval after the screenings if the tumor volume is higher than a certain proportion (taken as the 80% here) of the initial volume. The time periods when the doses are applied are indicated by a yellow background.
(PDF)

**S9 Fig. Results of virtual clinical trials comparing ID14 with their adaptive counterpart for different values of the decision threshold.** Kaplan-Meier curves showing the distribution of overall survival in virtual LGG patients undergoing TMZ treatment with ID14 protocols (individual doses given every 14 days) and their adaptive version (treatment interruption when the volume in 90-days periodic screenings is below a certain fraction of the volume at diagnosis). Six different trials were simulated for different values of the threshold deciding the administration of the treatment between 30% and 80%. The trials showed no significant difference between the survival of patients with the ID protocol or its adaptive version.
(PDF)

**S10 Fig. Results of virtual clinical trials comparing ID21 with their adaptive counterpart for different values of the decision threshold.** Kaplan-Meier curves showing the distribution of overall survival in virtual LGG patients undergoing TMZ treatment with ID14 protocols (individual doses given every 21 days) and their adaptive version (treatment interruption when the volume in 90-days periodic screenings is below a certain fraction of the volume at diagnosis). Six different trials were simulated for different values of the threshold deciding the administration of the treatment between 30% and 80%. The trials showed no significant difference between the survival of patients with the ID protocol or its adaptive version.
(PDF)

**S1 File. Patient population.** Characteristics, MRIs dates, longitudinal volumetric data and information related to treatment.
(XLSX)

**S2 File. Activation function $f(E)$.** How it was determined.
(ZIP)

**S3 File. Preliminary fits performed with the model.** In later fits, the value $\beta$ was fixed due to the proximity of this parameter to $\beta = 0.1$ day$^{-1}$ in all the studied cases. This file shows the results of these preliminary fits and the values obtained for the parameters.
(PDF)

## Author Contributions

**Conceptualization:** Thibault Delobel, Luis E. Ayala-Hernández, Jesús J. Bosque, Julián Pérez-Beteta, Víctor M. Pérez-García.

**Data curation:** Thibault Delobel, Luis E. Ayala-Hernández, Jesús J. Bosque, Julián Pérez-Beteta, Manuel García-Ferrer, Pilar Piñero, Philippe Schucht, Michael Murek, Víctor M. Pérez-García.

**Formal analysis:** Thibault Delobel, Luis E. Ayala-Hernández, Jesús J. Bosque, Salvador Chulián.

**Funding acquisition:** Víctor M. Pérez-García.

**Investigation:** Thibault Delobel, Luis E. Ayala-Hernández, Jesús J. Bosque, Julián Pérez-Beteta, Salvador Chulián.

**Methodology:** Thibault Delobel, Luis E. Ayala-Hernández, Jesús J. Bosque, Julián Pérez-Beteta, Salvador Chulián, Víctor M. Pérez-García.

**Project administration:** Jesús J. Bosque, Julián Pérez-Beteta, Víctor M. Pérez-García.

**Resources:** Julián Pérez-Beteta, Manuel García-Ferrer, Pilar Piñero, Philippe Schucht, Michael Murek.

**Software:** Thibault Delobel, Luis E. Ayala-Hernández, Jesús J. Bosque, Salvador Chulián.

**Supervision:** Jesús J. Bosque, Julián Pérez-Beteta, Víctor M. Pérez-García.

**Validation:** Thibault Delobel, Luis E. Ayala-Hernández, Jesús J. Bosque, Salvador Chulián.

**Visualization:** Thibault Delobel, Luis E. Ayala-Hernández, Jesús J. Bosque, Salvador Chulián.

**Writing – original draft:** Thibault Delobel, Luis E. Ayala-Hernández.

**Writing – review & editing:** Thibault Delobel, Luis E. Ayala-Hernández, Jesús J. Bosque, Julián Pérez-Beteta, Salvador Chulián, Manuel García-Ferrer, Pilar Piñero, Philippe Schucht, Michael Murek, Víctor M. Pérez-García.

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
