## [Decision Letter · Decision Letter 0]

13 Aug 2023

Dear Dr Bosque,

Thank you very much for submitting your manuscript "Overcoming chemotherapy resistances in low-grade gliomas: A computational approach" for consideration at PLOS Computational Biology.

As with all papers reviewed by the journal, your manuscript was reviewed by members of the editorial board and by several independent reviewers. In light of the reviews (below this email), we would like to invite the resubmission of a significantly-revised version that takes into account the reviewers' comments.

We cannot make any decision about publication until we have seen the revised manuscript and your response to the reviewers' comments. Your revised manuscript is also likely to be sent to reviewers for further evaluation.

Sincerely,

Jennifer A. Flegg

Academic Editor

PLOS Computational Biology

James O'Dwyer

Section Editor

PLOS Computational Biology

Reviewer's Responses to Questions

**Comments to the Authors:**

Reviewer #1: Review of the manuscript ‘Overcoming chemotherapy resistances in low-grade gliomas: A computational approach’ submitted to PLoS Computational Biology by Thibault Delobel, Luis E. Ayala-Hernández, Jesús J. Bosque, Julián Pérez-Beteta, Salvador Chulián, Manuel García-Ferrer, Pilar Piñero, Philippe Schucht, Michael Murek and Víctor M. Pérez-García

The present manuscript is a very detailed study of the mechanisms that lead low grade gliomas to resistance to the drug Temozolomide (TMZ), which, as far as I know, is the only drug able to pass the blood-brain barrier, and thus the drug on which one has to focus to understand pharmacoresistance in brain tumours. The study involves data recorded on patients, with their TMZ pharmacokinetic characteristics, and the construction of a population of virtual patients on these grounds. From the point of view of the study design, it is well documented and convincing.

However, I have real concerns about the ODE model, in particular as regards the subpopulation of resistant cells (V_R). It is not all clear to me, if this cell population is characterised by a strongly established resistance to TMZ, in a dead-end compartment, as it appears in the equations, whether it is due to an epigenetic (and thus in principle reversible, as in the case of EMT) change of phenotype (‘from proneural to mesenchymal transition’) or to a genetic mutation. If this resistant phenotype is fixed in such a way that no reversal is possible, by what mechanism is it fixed? If it is through an epimutation, then it should be reversible, be it only in the long term only. If it is through a genetic mutation, then equation (5) of the system makes sense, but such mutation should be searched for! By the way, the counter-intuitive (to me, if resistance is due to a costly non genetic mechanism, rather than to a thriftier established mechanism due to a forced - by long exposure to the drug - genetic mutation) mention of the fact that ρ_2 > ρ_1 would also make sense in the case of the second hypothesis.

The introduction of the intermediate compartments V_{PI} and V_P is natural and corresponds to the classical notions of drug-tolerant persisters and drug-tolerant expanded persisters, transposed from bacteriology to cancer biology in the princeps article of Sharma et al. in Cell 2010. The dead-end compartment V_R is something different from what is described in this princeps article. One could admit that strongly resistant cancer cells never die, being close to a stem cell state, but the unlimited exponential growth equation (5), even in the absence of stimulation by a drug, without any death nor fate reversal is intriguing, not to say that it is a big modelling problem.

Mentioning this princeps and fundamental article by Sharma et al., I must say that I am surprised that the authors do not include it in their references, which may explain why they qualify as ‘recently discovered’ the existence of persister cells, while the Sharma paper is thirteen years old. Of course, the contexts of lung cancer cells exposed to high doses of drugs in a Petri dish and that of low grade glioma cells in the brains of patients are different. However, speaking of drug tolerant persister cells in cancer, given their reversal to sensitivity as described by Sharma et al., this article cannot be ignored.

The fact that no reversal from the resistant cell population is prescribed in the model is to my meaning a bias of modelling which rules out the possibility of examining, with the help of these equations, adaptive therapy or so-called ‘drug holiday’ and ‘re-challenge’ strategies that have been proposed by teams of clinicians, whereas the possibility of reversal of drug tolerant cell populations to a sensitive state, that might be included in the model, would make it possible.

In summary, while I am favorably impressed by the sum of technical means presented to biologically and numerically investigate the question of resistance to TMZ in low grade gliomas, and to propose alternative therapeutical strategies, as compared to existing ones, I think that the ODE model is weak and highly questionable. To what extent it may fit or not to biological and clinical data cannot be an answer, not more than Ptolemy’s armillary sphere to justify pre-Copernican Earth-centred visions of the cosmos in astronomical calculations, right though they might have seemed, before the Copernican revolution. And if the ODE model presented in the present manuscript is not simply false, perhaps due to particularities of low grade gliomas exposed to temolozomide, then it must be fully biologically justified (and at least somewhat modified, for instance by at least putting a logistic or Gompertz-like growth term in equation (5)).

For these reasons, my evaluation is: to be resubmitted after major revision.

Reviewer #2: The review is uploaded as an attachment.

Reviewer #3: Review attached.

**Have the authors made all data and (if applicable) computational code underlying the findings in their manuscript fully available?**

Reviewer #1: Yes

Reviewer #2: **No: **Please provide the code of the simulations of ID treatments on virtual patients

Reviewer #3: **No: **The data is included in supplementary materials, but the code is not made available.

PLOS authors have the option to publish the peer review history of their article (what does this mean?). If published, this will include your full peer review and any attached files.

Reviewer #1: **Yes: **Jean Clairambault

Reviewer #2: No

Reviewer #3: No
---

## [Decision Letter · Decision Letter 1]

3 Nov 2023

Dear Dr Bosque,

We are pleased to inform you that your manuscript 'Overcoming chemotherapy resistance in low-grade gliomas: A computational approach' has been provisionally accepted for publication in PLOS Computational Biology.

Best regards,

Jennifer A. Flegg

Academic Editor

PLOS Computational Biology

James O'Dwyer

Section Editor

PLOS Computational Biology

Reviewer's Responses to Questions

**Comments to the Authors:**

Reviewer #1: The authors present convincing arguments, based on their particular data, to my main objection about the equation for TMZ-resistant GBM cells. Furthermore, they have agreed to my suggestion (this was mandatory to my eyes, as the Rabé paper of 2020 does not sum up all that has been found about persister cells in cancer) to add the Sharma et al. paper of 2010 to their bibliography. Now nothing opposes the publication of this very documented paper on TMZ resistance in GBM.

Reviewer #2: The authors have addressed all my comments and answered to all my questions in a very detailed and convincing way. The quality of the article greatly improved, and I think that it should be accepted and published.

Reviewer #3: The authors have taken into account suggested revisions and made improvements to the biological justification for the model, in particular the irreversible nature of resistance, clarified the limitations of the functional form used to model treatment, and strengthened the relevance of their clinical trial simulations. The revised version provides a well-documented, convincing model-based justification for testing alternative temozolomide protocols when treating low-grade gliomas.

**Have the authors made all data and (if applicable) computational code underlying the findings in their manuscript fully available?**

Reviewer #1: Yes

Reviewer #2: Yes

Reviewer #3: Yes

PLOS authors have the option to publish the peer review history of their article (what does this mean?). If published, this will include your full peer review and any attached files.

Reviewer #1: **Yes: **Jean Clairambault

Reviewer #2: No

Reviewer #3: No

---

## [Editor Report · Acceptance letter]

14 Nov 2023

PCOMPBIOL-D-23-00817R1 

Overcoming chemotherapy resistance in low-grade gliomas: A computational approach

Dear Dr Bosque,

I am pleased to inform you that your manuscript has been formally accepted for publication in PLOS Computational Biology. Your manuscript is now with our production department and you will be notified of the publication date in due course.

With kind regards,

Zsofia Freund
